behaviour, ecology, environmental science

bluefin tuna, California Current, energy landscape, heat increment of feeding, highly migratory species, migration

**Author for correspondence:**
Gemma Carroll
e-mail: gcarroll@edf.org

# Flexible use of a dynamic energy landscape buffers a marine predator against extreme climate variability

Gemma Carroll[1,2,3,4], Stephanie Brodie[1,2], Rebecca Whitlock[5], James Ganong[6], Steven J. Bograd[1,2], Elliott Hazen[1,2,6] and Barbara A. Block[6]

[1]Institute of Marine Science, University of California Santa Cruz, Santa Cruz, CA, USA
[2]Environmental Research Division, NOAA Southwest Fisheries Science Center, Monterey, CA, USA
[3]School of Aquatic and Fisheries Science, University of Washington, Seattle, WA, USA
[4]Environmental Defense Fund, San Francisco, CA, USA
[5]Department of Aquatic Resources, Swedish University of Agricultural Sciences, Drottningholm, Sweden
[6]Hopkins Marine Station, Stanford University, Monterey, CA, USA

GC, 0000-0001-7776-0946

Animal migrations track predictable seasonal patterns of resource availability and suitable thermal habitat. As climate change alters this 'energy landscape', some migratory species may struggle to adapt. We examined how climate variability influences movements, thermal habitat selection and energy intake by juvenile Pacific bluefin tuna (*Thunnus orientalis*) during seasonal foraging migrations in the California Current. We tracked 242 tuna across 15 years (2002–2016) with high-resolution archival tags, estimating their daily energy intake via abdominal warming associated with digestion (the 'heat increment of feeding'). The poleward extent of foraging migrations was flexible in response to climate variability, allowing tuna to track poleward displacements of thermal habitat where their standard metabolic rates were minimized. During a marine heatwave that saw temperature anomalies of up to +2.5°C in the California Current, spatially explicit energy intake by tuna was approximately 15% lower than average. However, by shifting their mean seasonal migration approximately 900 km poleward, tuna remained in waters within their optimal temperature range and increased their energy intake. Our findings illustrate how tradeoffs between physiology and prey availability structure migration in a highly mobile vertebrate, and suggest that flexible migration strategies can buffer animals against energetic costs associated with climate variability and change.

## 1. Introduction

Resource availability in the open ocean is patchily distributed and dynamic in space and time [1]. In response to this variability, many marine predators including whales, turtles, sharks, seabirds and pelagic fish have evolved highly migratory movement strategies. These can include basin-scale migrations between breeding and feeding grounds (e.g. [2]), as well as smaller scale 'foraging migrations' that track resources as they move on seasonal time-scales (e.g. [3,4]). Foraging migrations can consist of relatively rigid movement behaviours such as foraging site fidelity that exploit predictable productivity hotspots [5,6]), or migration routes that are optimized to match average patterns of resource availability [7]. Alternatively, animals may exhibit flexible movement strategies, using proximate cues to track resources as they become seasonally available across spatial gradients [8,9].

While there are benefits to optimizing foraging migrations to exploit known productive areas, there can be costs to strategies that rely on the predictability of resources [10]. Ocean systems are undergoing dramatic changes such as

warming trends [11,12] and increases in the frequency of marine heatwaves that cause similar magnitudes of disruption to ocean temperature over scales of weeks to years as century-scale climate change [13,14]. Such changes can alter the phenology of prey availability [15], and influence prey abundance and distributions through thermal disruption, increased stratification and declines in primary productivity [16,17]. If foraging migrations are hard-wired to specific locations, or if animals follow cues to productivity that become unreliable under novel climate conditions, rigid movement strategies could become maladaptive and result in ecological traps [7,18,19].

Mobile marine vertebrates must strike a balance between the need to find prey, and the need to minimize energetic costs associated with locomotion, foraging and thermoregulation. Migrations can therefore reflect a need to remain within an ambient temperature range that minimizes energy expenditure and maximizes physiological performance. This could constrain the ability of animals to migrate to areas of high prey availability that fall outside a narrow thermal range [20,21]. Environmental factors affecting energy expenditure via temperature effects on metabolic performance, and those affecting energy intake via prey availability, thus structure a complex 'energy landscape'. Tradeoffs in how animals optimize their use of this energy landscape remain poorly understood, despite this being essential to predicting their responses to environmental change.

Pacific bluefin tuna (*Thunnus orientalis*) are large pelagic fish that can reach sizes of up to 3 m and 450 kg at maturity. This species is highly mobile and can undertake multiple trans-oceanic migrations during their lifetimes between breeding grounds off the coast of Japan, and productive feeding grounds in the California Current system [22–24]. As juveniles and sub-adults (1–7 years old), a subset of the population enters the California Current where they remain resident for many years, undertaking seasonal north–south foraging migrations [3]. Pacific bluefin tuna have an extremely high commercial value and have been the target of intensive fishing efforts that have depleted the stock to only 4.5% of its historical size, necessitating their management at restricted quotas to support their recovery [25]. Understanding their seasonal movements in response to climate variability and change is critical to accurately assess the climate vulnerability of this species, and to inform climate-ready fisheries management efforts.

Here, we use biologging to assess how seasonal phenology and climate variability influenced the latitudinal migration extent, energy intake and thermal habitat selection of Pacific bluefin tuna over 15 years (2002–2016). Biologging is a powerful tool to remotely observe animal movement, physiology and foraging success [26–28]. Increasingly, long-term tagging programmes provide an opportunity to examine behaviour and migration in response to ecosystem dynamics and climate change, a crucial step towards understanding species' climate vulnerability and informing management outcomes [29,30].

We tracked migrations of Pacific bluefin tuna in the California Current with geolocating archival tags and sensors that measured pressure and ambient temperature. As Pacific bluefin tuna are regional endotherms that conserve metabolic heat within body tissues including their viscera [31], we also used sensors implanted in the visceral cavity to measure metabolic heat generated during digestion (i.e. the 'heat

increment of feeding'), from which we estimated daily energy intake [32–35]. We examine how climate affects the migration behaviour and energy landscape of Pacific bluefin tuna, and highlight the tunas' migratory and physiological responses to a marine heatwave that resulted in temperature anomalies of up to +6°C in the California Current, triggering significant declines in ecosystem productivity [14,36]. By assessing how migration pathways and energy intake of Pacific bluefin tuna change in response to a dynamic environment, we shed light on how flexible migratory foraging strategies can buffer highly mobile animals against potential energetic costs of shifting energy landscapes associated with climate variability in the Anthropocene.

## 2. Methods

### (a) Tagging

Pacific bluefin tuna ($n = 747$) were captured on hook and line during cruises aboard the F/V *Shogun* off California, USA, and Baja California, Mexico, between 2002 and 2016 (see [22,37] for fishing methods, surgery and release protocols). Bluefin tuna were surgically implanted with archival tags (Lotek, LTD 2310 series A–D) that measured ambient water temperature, peritoneal temperature, light and depth at intervals ranging from 4 to 120 s. Of the 747 tagged Pacific bluefin tuna, 242 were recaptured and used in this study. For these fish, we discarded the first 6 days of data because some fish were observed not to feed for several days after release [35]. We also removed data from the day fish were recaptured. The 242 recaptured Pacific bluefin tuna ranged between 62 and 156 cm in curved fork length (mean = $96.58 \pm 0.08$ cm; electronic supplementary material, figure S1), indicating that they were juveniles between 1 and 5 years old at release [38]. The number of recaptured individuals carrying tags in each year was variable, ranging from 2 in 2010, to 87 in 2003 (mean = $31 \pm 6.47$; electronic supplementary material, figure S2). Due to the commencement of the tagging program in mid-2002, data were not available in the first part of that year. Low tagging effort in the boreal summer 2013 meant that 2013 and 2014 had data from tagged fish available for only the first half and second halves of the year, respectively. The length of time that each fish was at large was variable, ranging from 1 to 876 days (mean = $289.5 \pm 11.09$ days).

### (b) Heat increment of feeding

We estimated daily energy intake by Pacific bluefin tuna using the heat increment of feeding (HIF), a measure of the heat produced in the visceral cavity during digestion [32]. For each Pacific bluefin tuna, we calculated daily HIF from a laboratory-calibrated algorithm adapted for the field that provided a time-integrated magnitude across which the bluefin tuna's peritoneal temperature was raised above a daily baseline representing the digestive system at rest (HIF area). For each wild fish, we measured daily HIF area and sea surface temperature (SST; the average tag-measured ambient temperature in the top 3 m of the water column over a 24 h period). These measured data were input into a hierarchical Bayesian regression model that was parameterized based on laboratory experiments with similar-sized captive Pacific bluefin tuna measured at a range of ambient temperatures (15–22°C), in order to relate the magnitude of daily HIF to the known energy density of ingested prey rations [34]. Daily energy intake of wild tuna (kcal d$^{-1}$) was estimated as the median of the posterior predictive distribution for a new Pacific bluefin tuna (i.e. one for which no experimental observations have been made), given daily observed HIF area and SST values. Here, we present values of daily energy intake

estimated from the model for a diet composed of Pacific sardines (*Sardinops sagax*), or forage fish of similar nutritional composition (see [34,35] for detailed descriptions of laboratory experiments, model development and application to wild juvenile Pacific bluefin tuna).

### (c) Location estimates

We estimated daily positions of tuna using light-based geolocation, with improvements to latitude estimates made by measuring overlap between tag-observed temperatures recorded within 1 m of the surface and satellite-observed SST [3,39]. Locations were processed using a Bayesian state-space model [40] to account for observation error and to interpolate missing location observations. Our aim was to understand variability in energy intake during feeding events in the California Current system, where most juvenile bluefin tuna remain resident for 1–7 years [37,41]. Although some juvenile Pacific bluefin tuna make occasional large forays into the central Pacific Ocean (electronic supplementary material, figure S3), the experiences of these individuals likely reflect different oceanographic and ecological processes from those affecting tuna when they are resident in the California Current. We therefore excluded daily positions outside the bounds of the California Current large marine ecosystem from our analysis (2% of total daily locations representing 25% of individuals).

### (d) Climate indices

We downloaded satellite-derived SST (UK Met Office Global Ocean OSTIA Sea Surface Temperature analysis product) at a native 0.05° × 0.05° daily resolution from https://marine.copernicus.eu and extracted daily values at the cell centroids of a 1° × 1° spatial grid covering the extent of the California Current large marine ecosystem ($n = 219$ cells). We took a mean value of SST across the California Current for each day of each year, as a system-wide index of oceanographic conditions. We then calculated the anomaly of these daily values from the 15-year mean for that day (figure 1a). For analyses exploring the effects of SST anomalies on migration behaviour and energy intake, we categorized each day according to whether it was experiencing a positive (warm) or negative (cool) anomaly.

### (e) Variation in latitudinal migration extent

To estimate interannual variability in foraging migration extent by Pacific bluefin tuna, we calculated the mean latitudinal position of all tuna on each calendar day. To reduce high-frequency variability and error associated with using daily values, we calculated a rolling 7-day mean. We calculated the coefficient of variation and the mean of the distance matrix for latitudinal position on each calendar day across all 15 years as indices of the magnitude of interannual variability.

### (f) Energy landscape

We calculated the mean satellite-derived SST in each 1° latitudinal bin in the California Current large marine ecosystem across all 15 years as a broad representation of spatial climate variability. To determine how ambient temperatures experienced *in situ* by Pacific bluefin tuna differed from these yearly averages, we calculated the mean ambient temperature recorded by tags when tuna were at each latitude in each year. We interpreted thermal habitat selection by Pacific bluefin tuna according to whether surface waters were below, within or above their metabolic minimum zone of 15–20°C, the preferred temperature range of this species within which standard metabolic rate is lowest [42]. To examine the spatial distribution of energy intake in the

California Current, we calculated the mean HIF-estimated daily energy intake in the same latitudinal bins.

### (g) Climate-driven variability in migration and energy intake

We used generalized additive models to estimate smoothed response curves of mean latitude and mean energy intake across the tunas' annual migration cycle as a function of SST anomaly (whether the California Current was warm or cool on that day). To determine whether tuna remained within their preferred temperature range of 15–20°C under both warm and cool conditions, we also estimated seasonal patterns of ambient temperature recorded by tags in relation to SST anomaly.

### (h) Response to marine heatwave

To understand how extreme climate events can structure patterns of energy intake and migration by Pacific bluefin tuna, we highlight 2015, the peak of a marine heatwave in the California Current. We calculated the mean spatial anomaly of energy intake by tuna in 2015 by subtracting mean values of energy intake in each 1° × 1° grid cell (all years except 2015) from the mean energy intake in that cell in 2015. We calculated the mean latitude at which tuna were foraging on each calendar day in 2015 and compared this to the long-term mean for that day. We then examined relationships between the mean daily foraging latitude of tuna and their daily energy intake anomaly, to determine whether variability in energy intake in 2015 was related to variability in their foraging migration path. Finally, we determined the proportion of time that tuna spent below, within and above their metabolic minimum zone of 15–20°C in 2015 compared to the mean calculated across all other years, as an indication of whether they were able to migrate flexibly to maintain a constant thermal experience under heatwave conditions.

Tag processing and HIF analyses were conducted in MATLAB v. 9.3.0 [43]. All other analyses were performed in R v. 3.6.3 [44]. Values are mean ± s.e. unless otherwise stated.

## 3. Results

### (a) Climate variability

The California Current system exhibited significant ocean climate variability during the study period (figure 1a). For example, during the 2014–2016, marine heatwave almost every day of the year was warmer than the 15-year average. In 2015, at the peak of the marine heatwave, there were days when SST across the entire region was more than 2.5°C warmer than the daily average, variability that is in line with expectations of century-scale climate change in this system [45].

### (b) Variation in latitudinal migration extent

We recovered archival tags from 242 juvenile Pacific bluefin tuna providing 68 507 daily locations and estimates of daily energy intake (electronic supplementary material, figure S3). The tunas have a north-south seasonal cycle of migration in the California Current, with their mean latitudinal migration extending from Baja California, Mexico (approx. 27.5° N) in late spring (May), to north of the Southern California Bight, USA (approx. 33° N) in autumn (October; figure 1b). Individual tuna swam as far south as approximately 20° N (south of the Baja California peninsula), and as far north as approximately 47° N (Washington, USA; see electronic supplementary

Proc. R. Soc. B 288: 20210671

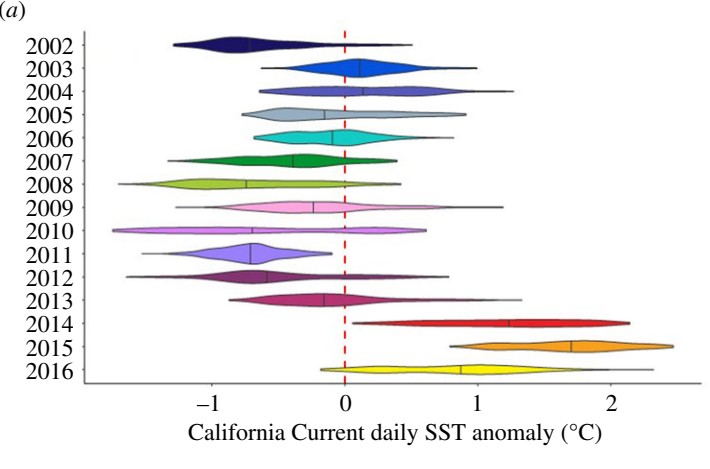

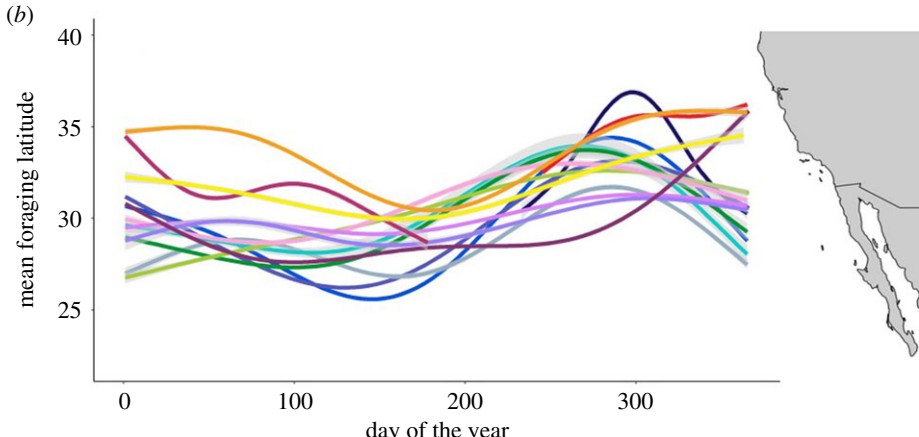

**Figure 1.** (*a*) distribution of average daily satellite-derived SSTs across the California Current from 2002 to 2016, expressed as anomalies from the 15-year average for that calendar day; (*b*) Mean daily foraging latitude of 242 juvenile Pacific bluefin tuna on each calendar day from 2002 to 2016. (Online version in colour.)

material, figure S4 for the variability of individual location estimates around annual trends). The average pairwise difference in mean daily latitudinal positions across all 15 years (matched by calendar day) was 2.31 ± 0.02° (approx. 250 km), and there was a mean interannual coefficient of variation between the same calendar day across years of 6.64 ± 0.07%.

There was a positive effect of fish size on the northern extent of Pacific bluefin tuna migrations (i.e. bigger fish generally swam further north), but tuna swam further north in warm years regardless of their size (electronic supplementary material, figure S5). Years when there were gaps in the time series (2002, 2013 and 2014) and years when tagged fish were unusually small relative to size distributions in other years (2011, 2012 and 2013) did not appear to produce effects that bias inferences about migration extent in response to climate variability (figures 1*b* and 2*a,b*). For example, despite being below average in size relative to size distributions over 15 years, tuna in 2013 and 2015 migrated further north on average than almost all other years during parts of the year when data were available (figure 1*b*).

### (c) Energy landscape

There was always a north-south gradient in mean annual satellite-derived SST in the California Current (figure 2*a*), but the average positions of specific thermal habitat thresholds were variable across years. For example, the mean location of the lower limit of the Pacific bluefin tunas' thermal minimum zone (15°C) differed by up to 8° (approx. 900 km) between years, which is likely to shape interannual

differences in the ability of tuna to access more northerly waters. Waters to the north and far south of the California Current had the most variable SSTs within years, while waters between 25° and 35° N most consistently provided optimal thermal habitat (electronic supplementary material, figure S6*a*). North-south gradients of ambient temperature exposure were also experienced by tuna (measured via archival temperature tags; figure 2*b*), but were most pronounced in the earlier parts of the study period (2002–2007). Across all years, average temperatures experienced by tuna at most foraging latitudes fell within their metabolic minimum zone of 15–20°C, indicating that they tracked their preferred thermal habitat. Tuna experienced the greatest variability in ambient temperature in the middle part of their migration each year, likely because they moved through these latitudes twice, in both spring and autumn (electronic supplementary material, figure S6*b*). Mean daily energy intake by juvenile Pacific bluefin tuna was highest in northern waters above 35°N, and this north-south gradient in energy intake persisted in most of the 15 years (figure 2*c*). Regions with the highest mean energy intake were often the most variable, but in some years, the highest latitudes saw the lowest variability (electronic supplementary material, figure S6*c*).

### (d) Climate-driven variability in migration and energy intake

Pacific bluefin tuna deviated from their average seasonal migration pathway in response to broad-scale SST anomalies

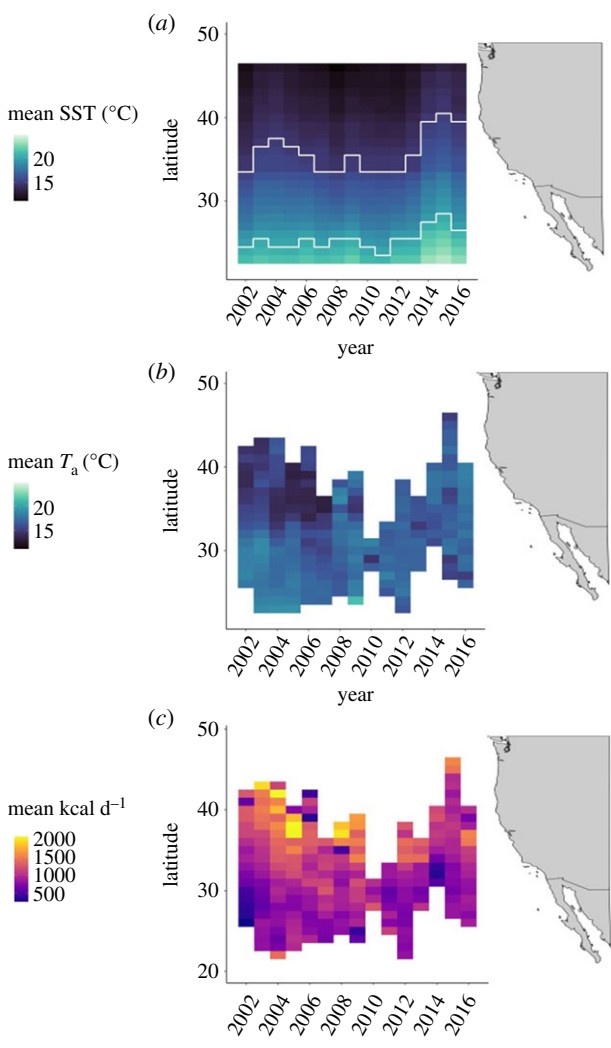

**Figure 2.** Components of the 'energy landscape' of juvenile Pacific bluefin tuna from 2002 to 2016: (a) mean satellite-derived SST in each latitude bin and year, with the locations of the lower (15°C) and upper (20°C) limits of the Pacific bluefin tuna's metabolic minimum zone within which standard metabolic rate is minimized shown as white lines; (b) ambient temperatures actually experienced by juvenile Pacific bluefin tuna in each latitude bin during their foraging migrations between 2002 and 2016; (c) mean energy intake (kcal d$^{-1}$) by juvenile Pacific bluefin tuna in each latitude bin during their foraging migrations between 2002 and 2016. (Online version in colour.)

in the California Current. Tuna were $0.88 \pm 0.04°$ (approx. 98 km) further north than usual for a given calendar day under warm water anomalies, and $0.55 \pm 0.03°$ (approx. 61 km) further south under cool anomalies (figure 3a). While this suggests that north-south movements may primarily serve to maintain an ambient temperature within their thermal minimum zone, tuna experienced some minor seasonal variability in thermal experience, with the highest ambient temperatures experienced during mid-summer (figure 3b). Ambient temperatures experienced by tuna were $0.16 \pm 0.02°C$ warmer than average under warm SST anomalies, and $0.19 \pm 0.02°C$ cooler under cool anomalies (figure 3b), though average values fell within the tunas' 15–20°C thermal minimum zone and were unlikely to be biologically significant. Tuna had slightly elevated seasonal patterns of energy intake under anomalously warm conditions (figure 3c), consuming an average of $34.36 \pm 3.29$ kcal d$^{-1}$ more than the overall mean, compared with

$32.82 \pm 3.41$ kcal d$^{-1}$ less when conditions were anomalously cold. They had a higher average energy intake in northern waters, with tuna consuming $90.82 \pm 3.74$ more kcal per day when they foraged further north than their mean foraging path, and $79.53 \pm 3.26$ fewer kcal when they foraged further south (figure 3d).

## (e) Response to marine heatwave

During the marine heatwave in 2015, spatially explicit estimates of energy intake were generally lower than average in the California Current, with the mean spatial anomaly across the California Current being $-145.93$ kcal d$^{-1} \pm 24.21$ ($n = 4697$ data points from 20 tagged tuna; figure 4a,b). Pacific bluefin tuna foraged further north on average throughout 2015, with a maximum northward anomaly of 8.09° (approx. 900 km) in March, when the tuna were foraging at a mean latitude of 35.23° N (figure 5a). An individual tuna foraged at a maximum latitude of 46.90° N, the most northerly foraging movement during the time series, approximately 1550 km further north than average for that day. While energy intake was slightly lower than average for a given day of the year at lower latitudes in 2015, tuna did better than the long-term average when they foraged at higher mean latitudes (figure 5b). Despite the California Current system being 2.5°C warmer on average during the heatwave in 2015 [46], tuna spent 1.5× more time foraging in cool waters below the lower limit (15°C) of their metabolic minimum zone (24% of time in 2015 cf. 16% over the whole study; figure 5c).

## 4. Discussion

Measures of foraging success derived from biologging sensors can augment studies of animal movement ecology to provide information on how the quality of the foraging landscape changes in relation to climate variability (e.g. [19,47]). By implanting temperature sensors in the visceral cavity of Pacific bluefin tuna, we were able to estimate their daily energy intake using well-tested associations between metabolic warming during digestion and the caloric value of prey rations [32–34,48]. Coupled with remotely sensed and *in situ* measurements of the tunas' thermal environment, we present new insights into how climate variability influences migration ecology across a changing energy landscape in a highly dynamic ocean system, the California Current.

We found that Pacific bluefin tuna foraging migrations were highly responsive to anomalous climate conditions. Deviations from the average migration path of approximately $-60$ and $+100$ km occurred when conditions across the California Current were anomalously warm or cold respectively, and one tuna foraged up to 1500 km further north than average in response to warm water anomalies during the 2015 marine heatwave. This outpaces even the extreme displacement of thermal habitat observed during this time [14]. This migratory flexibility served as an effective energetic strategy, optimizing the amount of time tuna remained within the 15–20°C zone where standard metabolic rate is minimized [42] and enabling them to increase their energy intake when local forage conditions were anomalously poor. Adaptively managing tradeoffs between prey consumption and metabolic costs associated with high temperatures is increasingly important in systems undergoing rapid climate change, where thermoregulatory costs are increasing [11,13] while productivity is declining [16].

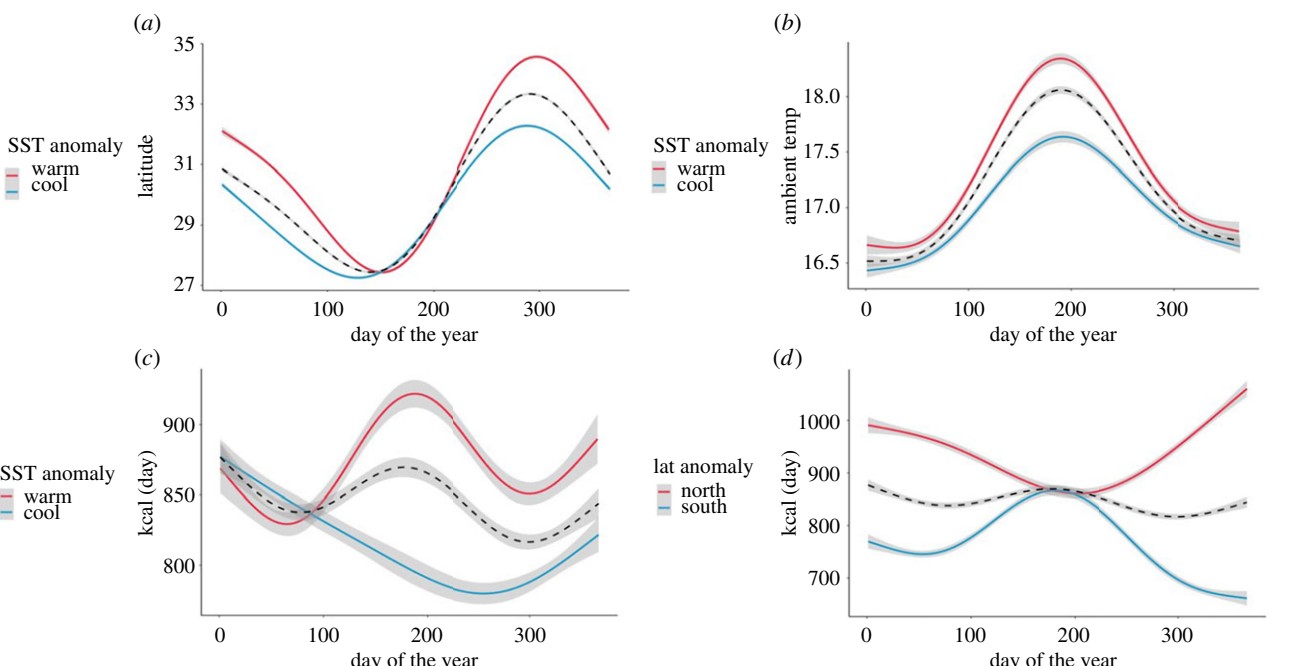

**Figure 3.** Effects of climate anomalies on foraging migration extent and energy intake by Pacific bluefin tuna in the California Current: (*a*) mean foraging latitude of Pacific bluefin tuna on each day of the year according to whether the California Current system was warmer (red) or cooler (blue) than the 15-year average. The mean trend for all individuals is shown by the black dashed line; (*b*) mean ambient temperature experienced by Pacific bluefin tuna according to whether the California Current system was warmer (red) or cooler (blue) than the 15-year average for that day; (*c*) mean energy intake by Pacific bluefin tuna according to whether the California Current system was warmer (red) or cooler (blue) than the 15-year average for that day; (*d*) mean energy intake by Pacific bluefin tuna according to whether they were further north (red) or south (blue) compared to the 15-year average for that day.

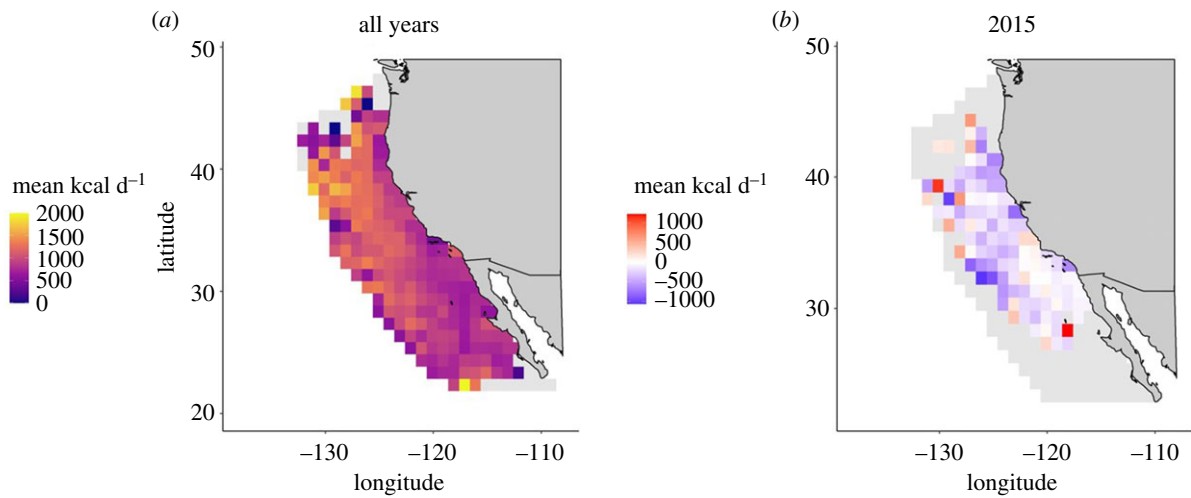

**Figure 4.** (*a*) Mean spatially explicit energy intake by juvenile Pacific bluefin tuna across the California Current large marine ecosystem over 15 years (2002–2016). (*b*) Mean spatially explicit energy intake anomalies by juvenile Pacific bluefin tuna in the California Current during a marine heatwave in 2015, relative to the average of all other tagging years. (Online version in colour.)

Although thermal habitat selection appears to be a principal driver of migration in bluefin tuna, we showed that tuna did not migrate flexibly with the sole purpose of remaining within their metabolic minimum zone. Tuna spent almost twice as many periods in 'cool' waters below this zone, than in 'warm' waters above this zone (figure 5*c*), indicating an active behavioural tradeoff between energetic costs and benefits conferred by foraging in cooler, more productive waters. This parallels observations of fine-scale behavioural strategies by albacore tuna, where individuals stay on the warm side of a front to maintain an elevated body temperature, but make regular forays into the cold side, to maximize prey capture and thus energy intake [49]. Unexpectedly, bluefin tuna in our study spent 1.5× more time below their metabolic minimum zone under heatwave conditions in 2015. This corresponds to periods when they accessed relatively cool areas in the north of the California Current, where they also increased their energy intake (figure 5*b*). Studies have suggested that compressed pockets of cool, high-quality foraging habitat remained available in parts of the California Current during the heatwave, aggregating prey and providing increased foraging opportunities for mobile species [50]. Our findings support these observations, with tuna remaining within low ambient temperatures even when they foraged at southern latitudes, despite surface temperatures being anomalously high in the system overall (figures 2*b* and 3*b*).

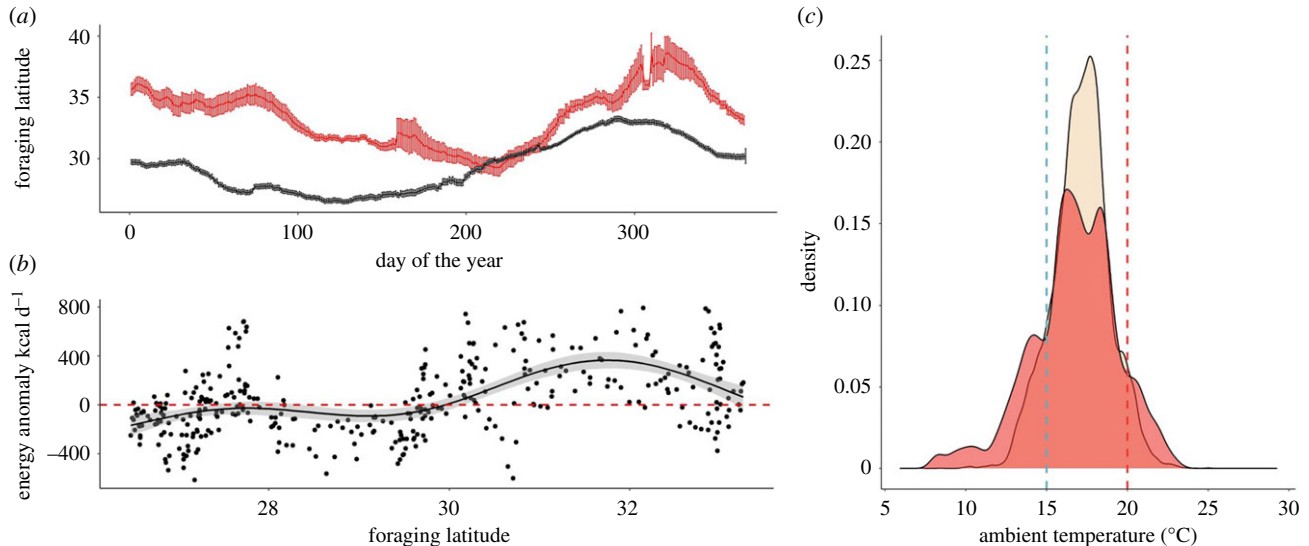

**Figure 5.** (a) Mean latitudinal migration extent by juvenile Pacific bluefin tuna in 2015 (red) compared to the mean in all other years (black); (b) mean daily energy intake anomaly by juvenile Pacific bluefin tuna in 2015 according to the mean latitude at which tuna were foraging on that day; (c) thermal habitat selection by juvenile Pacific bluefin tuna in 2015 (red) compared to the long-term mean (excluding 2015; beige). The locations of the lower (15°C, blue) and upper (20°C, red) bounds of the tunas' metabolic minimum zone are indicated by dashed lines.

The negative spatial energy intake anomalies that Pacific bluefin tuna experienced in 2015 suggest that the quality of the foraging landscape was generally reduced in the California Current during the heatwave (figure 4a), in line with observations of reduced primary productivity, and low forage fish and krill abundance during this period [36]. Tuna consumed relatively more energy than average only at times when they migrated unusually far north (figures 2c, 3d and 4b). Declines in local prey availability or quality, coupled with a lack of migratory flexibility, appear to explain why some top predators including sea lions, grey whales and some seabirds experienced starvation-induced mass mortalities in the California Current in the years immediately following this event [51–54]. Future work could complement our analyses of variation in the tunas' energy landscape by predicting the distributions of important prey species in relation to climate variability. This would provide independent measures of changes in the availability and quality of the forage base and allow us to assess how effectively the tuna were able to track changes in the availability of their prey.

Comprehensive diet analyses on Pacific bluefin tuna throughout the California Current would also help to resolve some of the uncertainty inherent in using estimates of energy intake based on the HIF, which can be sensitive to factors including the type of prey being digested [34,35]. For example, while we use equations to estimate energy intake based on laboratory experiments in which juvenile tuna were fed sardines [34], Pacific bluefin tuna can be dietary generalists when these high energy fish are not available. Their diet in southern and central California contains variable yet significant proportions of squid and pelagic red crab (*Pleuroncodes planipes* [55]), with the latter becoming a prominent part of tuna diet during 2014–2016, when these pelagic crabs experienced a range expansion into central Californian waters [56]. While the diet of bluefin tuna is likely to contain fewer squid and crustaceans and relatively more forage fish in the northern parts of their distribution, there remains substantial uncertainty in our understanding of shifts in predator–prey interactions that can only be resolved with a more comprehensive diet data collection throughout their latitudinal range.

Our findings that juvenile Pacific bluefin tuna migrate flexibly in response to climate variability and maintain a high energy intake under anomalously warm ocean conditions have important implications for managing this population. Bluefin tuna fisheries operating in the California Current have seen a recent rise in catches, including during the 2014–2016 marine heatwave [57]. These increases in catch rates do not appear to be related to an increase in stock biomass, which remains at near-historic lows with less than 5% of the population remaining [24]. Instead, these changes appear to represent an increase in the presence of larger bluefin tuna in the California Current potentially due to changes in the distribution of suitable habitat or the availability of key prey species [57]. Climate change is expected to increase temperatures and decrease productivity in much of the California Current [45], potentially leading to shifts in the seasonal migrations of juvenile Pacific bluefin tuna which may affect their availability to commercial and recreational fisheries in Mexican and US waters. It remains an important question how changes in resource availability in their juvenile foraging grounds might influence the overall fitness and structure of the population in the North Pacific, by affecting the timing of basin-scale migrations to breeding grounds in the western Pacific, or the proportion of the population that makes these migrations each year. Understanding how climate variability and change alter patterns of individual migrations and ultimately drive population-level shifts in species distributions is essential to implementing climate-ready management of highly migratory marine animals like tunas.

Ethics. All animal research was conducted in accordance with IACUC protocols from Stanford University.

Data accessibility. Data and code from which analyses and figures presented in this paper can be reproduced are available on GitHub (github.com/gemcarroll/bluefin). Raw data from sensors deployed on bluefin tuna are available from www.gtopp.org.

Authors' contributions. G.C.: conceptualization, formal analysis, investigation, methodology, visualization and writing-original draft; S.B.: conceptualization, investigation, writing-review and editing; R.W.: formal analysis, methodology, software, writing-review and editing;

J.G.: data curation, software, writing-review and editing; S.J.B.: conceptualization, supervision, writing-review and editing; E.H.: conceptualization, supervision, writing-review and editing; B.A.B.: conceptualization, data curation, funding acquisition, investigation, methodology, project administration, resources, supervision, writing-review and editing.

All authors gave final approval for publication and agreed to be held accountable for the work performed therein.

Competing interests. We declare we have no competing interests.

Funding. Funding for this work was received through the Tagging of Pacific Pelagics program from the Moore, Packard, and Monterey Bay Aquarium Foundations, the Tag-a-Giant Fund of The Ocean Foundation, and from the Office of Naval Research. Personnel support was provided by Stanford University, the University of California Santa Cruz and NOAA's Southwest Fisheries Science Center.

Acknowledgements. The authors thank the captains T. Dunn, N. Kagawa, B. Smith, A. Barnhill and crew of the fishing vessel *Shogun* for their dedication and help with capture, archival tagging and release of Pacific bluefin tuna. We thank the technical staff of the Tuna Research and Conservation Center (R. Schallert, J. Noguiera, L. Rodriguez, and Stanford undergraduate and graduate students and postdocs), A. Norton, C. Farwell and the many individuals that helped in tagging efforts through the 15 years of cruises. The authors are grateful to the Mexican government for permitting access to Pacific bluefin tunas in Mexican waters across the study years and to O. Sosa-Nishizaki of the Center for Scientific Research and Higher Education in Ensenada (CICESE) and T. Baumgartner McBride, Department of Biological Oceanography, Centro de Investigación CientÍfica y de Educación Superior de Ensenada University, for his recent support. Thanks to M. Castleton for technical assistance, and Dr B. Muhling for helpful comments on an earlier version of this manuscript.

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
