## [Peer Review File · Proceedings of the Royal Society B: Biological Sciences]

Review History

RSPB-2021-0671.R0 (Original submission)

Review form: Reviewer 1

Recommendation

Accept with minor revision (please list in comments)

Scientific importance: Is the manuscript an original and important contribution to its field?

Good

General interest: Is the paper of sufficient general interest?

Good

Quality of the paper: Is the overall quality of the paper suitable?

Good

Is the length of the paper justified?

Yes

Should the paper be seen by a specialist statistical reviewer?

No

Do you have any concerns about statistical analyses in this paper? If so, please specify them explicitly in your report.

Yes

It is a condition of publication that authors make their supporting data, code and materials available - either as supplementary material or hosted in an external repository. Please rate, if applicable, the supporting data on the following criteria.

Is it accessible?

Yes

Is it clear?

Yes

Is it adequate?

Yes

Do you have any ethical concerns with this paper?

No

Comments to the Author

Overview

The authors use biologging data from 242 juvenile bluefin tuna, obtained over 15 years (2002-2006) to investigate how variability in sea surface temperature influences their distribution and energy intake. Energy intake is inferred from a model applied to temperature data on abdominal warming resulting from prey ingestion. Distribution is measured from light level geolocators with sea surface temperature correction of the location estimates. The authors examine how these tuna shift their distribution within and across years in response to variation in sea surface temperatures, as they counterbalance the requirement to remain in certain thermal conditions with the need to find prey. Special attention is given to 2015, an anomalously warm year. They show that tuna distribution is flexible in response to sea surface temperatures and discuss the implications this has for management of bluefin tuna.

The work uses an impressive dataset on a large number of individuals (although see my specific comment on the number of individuals per year) and the integration of physiological requirements with estimates of energy intake is a key strength.

I found the analyses to be well conducted and could identify no major problems. I appreciate that the authors have made their code available. Overall, the manuscript is well-written and neatly presented. The figures are clear and attractive.

In my opinion the results are interesting and relevant in terms of the ability of marine predators to respond flexibly to variations in environmental conditions, given that the ability of predators to respond to such variations will be an important factor in how they deal with environmental change. In contrast to lots of other work on, for example seals and seabirds, I found the thermal requirements of tuna to be an interesting element in this work which nicely illustrates the constraint of meeting physiological requirements in the environment while finding food.

Overall, I have not major criticisms or concerns, only one broad question and several minor questions and suggestions (below).

General questions/comments:

1. Is it a reasonable assumption that tuna are targeting the same or similar prey when their distribution is so different across time? That is, does the relationship between the measured 'heat increment of feeding' and the energy ingested hold when tuna change their distribution? I see mention of this in the discussion (with reference to Whitlock et al. 2013), but perhaps an

additional sentence or two on the implications for the results could be added.

Minor comments

L22: 'seasonal movements of prey resources'. To my mind this excludes the seasonal emergence and disappearance of resources (or changes in resource abundance), which may also drive migration (rather than simply being a case of prey resources moving). This statement also does not quite agree with one in the introduction (L59-61) where you do point out that movement strategies may be flexible, in response to more spatio-temporally dynamic (less predictable?) resource distributions.

L65: Replace 'seeing' with 'undergoing'.

L67: Give the typical timescale of these marine heatwaves so that the contrast with the later 'century-scale climate change' is more clear.

L71: Perhaps make it clear that, rather than the cues degrading, there is a breakdown between the cue and the condition/state, etc. the cue is signalling. Minor semantics, but when I read 'cues that degrade' I got the impression of weakening signals rather than poor or false signals (and I think the latter is the case for ecological traps). Maybe Sherley et al. (<https://doi.org/10.1016/j.cub.2016.12.054>) is a better example than Gremillet et al. (2008) (same Benguela system)?

L86: Perhaps a size/mass indication for 'large'? 'Large' would mean very different things to this journal's broad readership and I guess many of us do not know just how big tuna are!

L92: Rephrase for clarity, please. Do you mean here that the population level is 95.5% lower than what it was prior to fishing? Or that the proportion of the population that is unfished (not sure what that would mean) has decreased by 95.5%.

L98: While you have used tracking data, I would say that you (arguably) have not assessed their 'migration pathway' (a longitudinal approach), but only their spatial distribution (cross-sectional approach) at discrete times. Maybe 'spatio-temporal distribution' or something like that?

L117: Presumably climate variability and change are not only related with energetic costs for highly mobile animals? The winners and losers narrative is widespread. Maybe change to 'can buffer highly mobile animals against the shifting energyscape associated with...!'

L132: Looking at the size distribution of individuals across years (Figure S1), it seems that individuals in 2011-2013 were very small (young) compared to other years (explained, I imagine, by small sample sizes in at least some of those years). What kind of influence do you think this (and the very small sample sizes in some years [2010, 2013], generally) has on the results?

L160-161: Tag-observed SST also in surface waters as for the temperature average? And the same SST product used later in the analyses?

L168-170: How were these bounds determined? Given that your manuscript is about environmental variability over 15 years, was the boundary of the California Current LME dynamic over the study? From Figure S3 I see that some movements are clearly long-distance migration outside the California Current, but what about more borderline cases? Do you think they could be responses to dynamic conditions, just as the north-south movements you focus on are?

L173: 'migration path'. See my earlier comment.

L175-176: Does calculating this moving average not introduce much more non-independence in the data that are used in the GAMs and GLMs?

L176: I don't think I understand how the pairwise distances are calculated for 15 years. Are the pairwise distances for adjacent years, and then the mean is taken?

L260: This result highlights my earlier question about the small individuals in some years. Were mean latitudes biased downwards in those years because only small individuals were tracked? Did this pattern of larger individuals occurring further north hold within years, or did size confound the year-to-year variation in foraging latitude? Perhaps give the GLM results here if they speak to this question.

L263-264: See my earlier request to clarify how this pairwise distance is calculated.

L279-280: It would be useful to have the degree shifts as approximate km values too (as in the abstract).

L391: Delete 'fitness', since you do not assess the fitness of these individuals as usually defined. Energetic benefits?

Figure 1: From the truncation of the smooths in some years, I guess that in these years the data did not cover the whole year. How would this influence your results? Could the results from some years cause bias simply because they did not cover the whole year?

Figure S4: It seems that part A of this figure is missing (and part B is not labelled as such). Also, one can figure out which smooth response is which based on their relative colours, but it would be better to label or clearly colour the two lines.

Review form: Reviewer 2

Recommendation

Accept with minor revision (please list in comments)

Scientific importance: Is the manuscript an original and important contribution to its field?

Excellent

General interest: Is the paper of sufficient general interest?

Excellent

Quality of the paper: Is the overall quality of the paper suitable?

Excellent

Is the length of the paper justified?

Yes

Should the paper be seen by a specialist statistical reviewer?

No

Do you have any concerns about statistical analyses in this paper? If so, please specify them explicitly in your report.

No

It is a condition of publication that authors make their supporting data, code and materials available - either as supplementary material or hosted in an external repository. Please rate, if applicable, the supporting data on the following criteria.

Is it accessible?

No

Is it clear?

No

Is it adequate?

No

Do you have any ethical concerns with this paper?

No

Comments to the Author

Dynamic energy landscapes are critical in determining movement patterns of predators, yet incorporating energy intake and expenditure to determine such movement is challenging. The authors use energy intake (from tags) and expenditure (from temperature) to create a dynamic energy landscape in iconic bluefin tuna, and show how tuna alter their migration to optimize their landscape each year. I have only a few relatively small comments:

1. Most of the figures show modelled data using GAMMs. Thus, it's not possible to assess whether trends are driven by outliers, the goodness-of-fit or whether the non-linearity is appropriate. I encourage the authors to add the actual data, perhaps as additional panels, to figures 1 and 3.
2. In Figure 2, why the SE's? I would have thought the SD would be more appropriate. Otherwise, you are essentially calculating $1/\sqrt{\text{sample size}}$ and so representing the amount of data you collected rather than variability.
3. I do not believe the raw (location and other sensor) data is available. It is standard to make such data available via DRYAD or Movebank or similar.
4. The calibrations of body temperature as a measure of energy intake appear to be small-scale. Over such large scales could it be more difficult to detect body temperature drops in warm waters and so the lower energy intake is partially an artefact of lower body temperature drops in warm water?
5. L129. Why six days?

Decision letter (RSPB-2021-0671.R0)

04-May-2021

Dear Dr Carroll:

Your manuscript has now been peer reviewed and the reviews have been assessed by an Associate Editor. The reviewers' comments (not including confidential comments to the Editor) and the comments from the Associate Editor are included at the end of this email for your reference. As you will see, the reviewers and the Editors have raised some concerns with your manuscript and we would like to invite you to revise your manuscript to address them.

Research ethics:

Use of animals and field studies:

It is a condition of publication that you make available the data and research materials supporting the results in the article. Please see our Data Sharing Policies (<https://royalsociety.org/journals/authors/author-guidelines/#data>). Datasets should be deposited in an appropriate publicly available repository and details of the associated accession number, link or DOI to the datasets must be included in the Data Accessibility section of the article (<https://royalsociety.org/journals/ethics-policies/data-sharing-mining/>). Reference(s) to datasets should also be included in the reference list of the article with DOIs (where available).

Please submit a copy of your revised paper within three weeks. If we do not hear from you within this time your manuscript will be rejected. If you are unable to meet this deadline please let us know as soon as possible, as we may be able to grant a short extension.

Best wishes,
Dr Sasha Dall
mailto:proceedingsb@royalsociety.org

Associate Editor
Board Member: 1
Comments to Author:

We have obtained two reviews of your manuscript and both referees were positive about your work. I agree with the referees' assessment and believe this paper may be suitable for publication in Proceedings B. However, one of the reviewers provided several questions for consideration and suggestions for improvement that I concur with, primarily around clarifying some of the methods, figures, and using precise language. Further, clarification regarding how some missing data might have impacted your results would be good as well. For example, in Figure 1B it seems the year 2014 starts at day ~230 - does this mean that none of the animals tagged prior to that date were transmitting data? Is this because of the timing of the tagging efforts each year? Visualization-wise, I am also not sure of the combination of some of panels within the figures. For example, isn't Figure 5D already shown, effectively, in Figure 4A? Reviewing the figures to ensure each supports a point in the results would be useful.

While the other reviewer had fewer comments, a few with regard to the statistical analysis, I feel their concerns should be clarified and I think would improve the paper. Please note that one reviewer commented that the raw data were not available, but I was able to access "hiflocs.csv" via Github, so please do advise if this is the full dataset from which your models could be recreated. If there are other data that are required please be sure to include them in your Github repository.

Reviewer(s)' Comments to Author:
Referee: 1
Comments to the Author(s)
Overview

The authors use biologging data from 242 juvenile bluefin tuna, obtained over 15 years (2002-2006) to investigate how variability in sea surface temperature influences their distribution and

energy intake. Energy intake is inferred from a model applied to temperature data on abdominal warming resulting from prey ingestion. Distribution is measured from light level geolocators with sea surface temperature correction of the location estimates. The authors examine how these tuna shift their distribution within and across years in response to variation in sea surface temperatures, as they counterbalance the requirement to remain in certain thermal conditions with the need to find prey. Special attention is given to 2015, an anomalously warm year. They show that tuna distribution is flexible in response to sea surface temperatures and discuss the implications this has for management of bluefin tuna.

The work uses an impressive dataset on a large number of individuals (although see my specific comment on the number of individuals per year) and the integration of physiological requirements with estimates of energy intake is a key strength.

I found the analyses to be well conducted and could identify no major problems. I appreciate that the authors have made their code available. Overall, the manuscript is well-written and neatly presented. The figures are clear and attractive.

In my opinion the results are interesting and relevant in terms of the ability of marine predators to respond flexibly to variations in environmental conditions, given that the ability of predators to respond to such variations will be an important factor in how they deal with environmental change. In contrast to lots of other work on, for example seals and seabirds, I found the thermal requirements of tuna to be an interesting element in this work which nicely illustrates the constraint of meeting physiological requirements in the environment while finding food.

Overall, I have not major criticisms or concerns, only one broad question and several minor questions and suggestions (below).

General questions/comments:

1. Is it a reasonable assumption that tuna are targeting the same or similar prey when their distribution is so different across time? That is, does the relationship between the measured 'heat increment of feeding' and the energy ingested hold when tuna change their distribution? I see mention of this in the discussion (with reference to Whitlock et al. 2013), but perhaps an additional sentence or two on the implications for the results could be added.

Minor comments

L22: 'seasonal movements of prey resources'. To my mind this excludes the seasonal emergence and disappearance of resources (or changes in resource abundance), which may also drive migration (rather than simply being a case of prey resources moving). This statement also does not quite agree with one in the introduction (L59-61) where you do point out that movement strategies may be flexible, in response to more spatio-temporally dynamic (less predictable?) resource distributions.

L65: Replace 'seeing' with 'undergoing'.

L67: Give the typical timescale of these marine heatwaves so that the contrast with the later 'century-scale climate change' is more clear.

L71: Perhaps make it clear that, rather than the cues degrading, there is a breakdown between the cue and the condition/state, etc. the cue is signalling. Minor semantics, but when I read 'cues that degrade' I got the impression of weakening signals rather than poor or false signals (and I think the latter is the case for ecological traps). Maybe Sherley et al. (<https://doi.org/10.1016/j.cub.2016.12.054>) is a better example than Gremillet et al. (2008) (same Benguela system)?

L86: Perhaps a size/mass indication for 'large'? 'Large' would mean very different things to this journal's broad readership and I guess many of us do not know just how big tuna are!

L92: Rephrase for clarity, please. Do you mean here that the population level is 95.5% lower than what it was prior to fishing? Or that the proportion of the population that is unfished (not sure what that would mean) has decreased by 95.5%.

L98: While you have used tracking data, I would say that you (arguably) have not assessed their 'migration pathway' (a longitudinal approach), but only their spatial distribution (cross-sectional approach) at discrete times. Maybe 'spatio-temporal distribution' or something like that?

L117: Presumably climate variability and change are not only related with energetic costs for highly mobile animals? The winners and losers narrative is widespread. Maybe change to 'can buffer highly mobile animals against the shifting energyscape associated with...!'

L132: Looking at the size distribution of individuals across years (Figure S1), it seems that individuals in 2011-2013 were very small (young) compared to other years (explained, I imagine, by small sample sizes in at least some of those years). What kind of influence do you think this (and the very small sample sizes in some years [2010, 2013], generally) has on the results?

L160-161: Tag-observed SST also in surface waters as for the temperature average? And the same SST product used later in the analyses?

L168-170: How were these bounds determined? Given that your manuscript is about environmental variability over 15 years, was the boundary of the California Current LME dynamic over the study? From Figure S3 I see that some movements are clearly long-distance migration outside the California Current, but what about more borderline cases? Do you think they could be responses to dynamic conditions, just as the north-south movements you focus on are?

L173: 'migration path'. See my earlier comment.

L175-176: Does calculating this moving average not introduce much more non-independence in the data that are used in the GAMs and GLMs?

L176: I don't think I understand how the pairwise distances are calculated for 15 years. Are the pairwise distances for adjacent years, and then the mean is taken?

L260: This result highlights my earlier question about the small individuals in some years. Were mean latitudes biased downwards in those years because only small individuals were tracked? Did this pattern of larger individuals occurring further north hold within years, or did size confound the year-to-year variation in foraging latitude? Perhaps give the GLM results here if they speak to this question.

L263-264: See my earlier request to clarify how this pairwise distance is calculated.

L279-280: It would be useful to have the degree shifts as approximate km values too (as in the abstract).

L391: Delete 'fitness', since you do not assess the fitness of these individuals as usually defined. Energetic benefits?

Figure 1: From the truncation of the smooths in some years, I guess that in these years the data did not cover the whole year. How would this influence your results? Could the results from some years cause bias simply because they did not cover the whole year?

Figure S4: It seems that part A of this figure is missing (and part B is not labelled as such). Also, one can figure out which smooth response is which based on their relative colours, but it would be better to label or clearly colour the two lines.

Referee: 2

Comments to the Author(s)

Dynamic energy landscapes are critical in determining movement patterns of predators, yet incorporating energy intake and expenditure to determine such movement is challenging. The authors use energy intake (from tags) and expenditure (from temperature) to create a dynamic energy landscape in iconic bluefin tuna, and show how tuna alter their migration to optimize their landscape each year. I have only a few relatively small comments:

1. Most of the figures show modelled data using GAMMs. Thus, it's not possible to assess whether trends are driven by outliers, the goodness-of-fit or whether the non-linearity is appropriate. I encourage the authors to add the actual data, perhaps as additional panels, to figures 1 and 3.
2. In Figure 2, why the SE's? I would have thought the SD would be more appropriate. Otherwise, you are essentially calculating $1/\sqrt{\text{sample size}}$ and so representing the amount of data you collected rather than variability.
3. I do not believe the raw (location and other sensor) data is available. It is standard to make such data available via DRYAD or Movebank or similar.
4. The calibrations of body temperature as a measure of energy intake appear to be small-scale. Over such large scales could it be more difficult to detect body temperature drops in warm waters and so the lower energy intake is partially an artefact of lower body temperature drops in warm water?
5. L129. Why six days?

Author's Response to Decision Letter for (RSPB-2021-0671.R0

See Appendix A.

Decision letter (RSPB-2021-0671.R1)

21-Jun-2021

Dear Dr Carroll

I am pleased to inform you that your manuscript RSPB-2021-0671.R1 entitled "Flexible use of a dynamic energy landscape buffers a marine predator against extreme climate variability" has been accepted for publication in Proceedings B.

The referee(s) have recommended publication, but also suggest some minor revisions to your manuscript. Therefore, I invite you to respond to the referee(s)' comments and revise your manuscript. Because the schedule for publication is very tight, it is a condition of publication that

you submit the revised version of your manuscript within 7 days. If you do not think you will be able to meet this date please let us know.

[http://datadryad.org/submit?journalID=RSPB&manu=\(Document not available\)](http://datadryad.org/submit?journalID=RSPB&manu=(Document%20not%20available)) which will take you to your unique entry in the Dryad repository. If you have already submitted your data to dryad you can make any necessary revisions to your dataset by following the above link. Please see <https://royalsociety.org/journals/ethics-policies/data-sharing-mining/> for more details.

Sincerely,
Dr Sasha Dall
Editor, Proceedings B
<mailto:proceedingsb@royalsociety.org>

Associate Editor:

Comments to Author:

My only remaining concern/question is that the raw data is apparently only available via emailing Dr. Barbara Block, rather than being deposited on Dryad or Movebank or something similar. The authors suggest that everything can be replicated with the derived dataset that is provided on GitHub, but this is a question I wasn't sure about, and is my only remaining hesitation. For reference, I can understand this a little because I, too, have experience with datasets where the raw data are nearly meaningless and really need to be cleaned up to be useful, but on the other hand in my experience we showed (via GitHub) how we did that. Further to this point, given the dataset spans 15 years, I am wondering what the funding sources were that allowed this data collection? I did not see mention of funding in the Acknowledgements, unless I missed it somehow, and that is a little strange, I think? Anyway, this seems like a fairly minor detail that is easily cleaned up, but I think it would be good to understand prior to accepting for publication.

Author's Response to Decision Letter for (RSPB-2021-0671.R1)

See Appendix B.

Decision letter (RSPB-2021-0671.R2)

12-Jul-2021

Dear Dr Carroll

I am pleased to inform you that your manuscript entitled "Flexible use of a dynamic energy landscape buffers a marine predator against extreme climate variability" has been accepted for publication in Proceedings B.

Data Accessibility section

Open Access

Paper charges

Sincerely,

Dr Sasha Dall

Appendix A

Reviewer Comments	Response
Associate Editor We have obtained two reviews of your manuscript and both referees were positive about your work. I agree with the referees' assessment and believe this paper may be suitable for publication in Proceedings B. However, one of the reviewers provided several questions for consideration and suggestions for improvement that I concur with, primarily around clarifying some of the methods, figures, and using precise language.	We thank the Associate Editor and the two reviewers for their considered appraisal of our work. We are glad they find the content our manuscript suitable for publication in Proceedings B. We have now addressed the concerns laid out by the reviewers, including those that the Editor has highlighted in their comments to us. Specifically, we have addressed the issues with figures and terminology that Reviewer 1 identified, and we believe that the paper is now more precise both in terms of describing background ecological theory, and in describing the methods. ***Please note that line numbers refer to the track changes version of the manuscript at the end of this document***
Further, clarification regarding how some missing data might have impacted your results would be good as well. For example, in Figure 1B it seems the year 2014 starts at day ~230 - does this mean that none of the animals tagged prior to that date were transmitting data? Is this because of the timing of the tagging efforts each year?	Due to unavoidable variability in tagging effort in this 15-year field program, a couple of years had truncated data. Fish were usually tagged in the North American summer (mid-year), so the first year (2002) began at that time. When there was a year when tags were not deployed (2013), data from the 2nd half of that year and the 1st half of 2014 are 'missing'. Unfortunately we can't fill these gaps, but we present sample sizes and the distribution of data transparently in Figures 1, S2, and a new Figure S4 that we added based on a suggestion by Reviewer 2. We have now modified the methods to clarify where there are gaps in tagging effort and that these correspond to a period where there were no fish 'at large' in this dataset (Lines 130-134). Our analyses in this paper are aggregated seasonally (i.e. based on calendar day) and spatially, and we generally avoid making between-year comparisons (except for highlighting 2015 as an anomalous heatwave year). We have no reason to believe that missing data is biasing the results that we present. 2014 was a warm year at the start of the marine heatwave that peaked in 2015, and tuna were migrating further north than usual during the parts of the year when we did have tracks (Figure 1A, 2A, 2B). This is in line with our findings regarding the effect of

	SST anomaly on migration extent (e.g. Figure 3A). If the full years of data were available it's possible that our results would be strengthened further, but we don't feel it is possible to speculate. We have now explicitly discussed both data availability and variability in the size of tagged fish across years in response to these comments from the Editor and Reviewer 1 in the Results (Lines 276-281).
Visualization-wise, I am also not sure of the combination of some of panels within the figures. For example, isn't Figure 5D already shown, effectively, in Figure 4A? Reviewing the figures to ensure each supports a point in the results would be useful.	We thank the Editor for their advice regarding the figures and have taken this on board. We have removed panel 1B where most of the information was duplicated in other figures, and we have replaced it with panel A showing SST anomalies, in order to better allow readers to visualise climate variability in each year in relation to migration behaviour (in response to thoughtful questions by Reviewer 1). We moved the original figures 2C and 2D to supplement, and instead added a spatial representation of the thermal landscape (2A) over 15 years because this more closely addresses points in the results. We agree with the Editor that information in Figure 4A was duplicated in 5D, so we removed panel 4A which had less information. We have now separated the panels showing spatial energy intake anomalies in 2015 into their own figure, because we feel that this is an important result that should not be lost among other information (Figure 5). We feel that all the figures are now much simpler, clearer, and that each supports a unique point in the results. We believe that the clarity and impact of the manuscript is increased as a result.

While the other reviewer had fewer comments, a few with regard to the statistical analysis, I feel their concerns should be clarified and I think would improve the paper. Please note that one reviewer commented that the raw data were not available, but I was able to access "hiflocs.csv" via Github, so please do advise if this is the full dataset from which your models could be recreated. If there are other data that are required please be sure to include them in your Github repository.	We have addressed the comments that Reviewer 2 had pertaining to statistical recommendations and display of raw data (we have added a new Figure S4, and have changed display of standard error to standard deviation in Figures S6A, B and C). The data set in the GitHub repository (hiflocs.csv) is a processed dataset that is sufficient to conduct all the analyses in this paper (i.e. the lead author used these data to conduct all analyses). We have now added a note in the data accessibility statement that the raw data files are available upon request from Prof BA Block, but that all the necessary information (i.e. locations of the animals, information on their size, estimates of their daily energy intake, their internal and ambient temperatures, etc.) is all publicly accessible via GitHub, and the entire study can be reproduced from this dataset with the available code.
Reviewer 1	
The authors use biologging data from 242 juvenile bluefin tuna, obtained over 15 years (2002-2006) to investigate how variability in sea surface temperature influences their distribution and energy intake. Energy intake is inferred from a model applied to temperature data on abdominal warming resulting from prey ingestion. Distribution is measured from light level geolocators with sea surface temperature correction of the location estimates. The authors examine how these tuna shift their distribution within and across years in response to variation in sea surface temperatures, as they counterbalance the requirement to remain in certain thermal conditions with the need to find prey. Special attention is given to 2015, an anomalously warm year. They show that tuna distribution is flexible in response to sea surface temperatures and discuss the implications this has for management of bluefin tuna. The work uses an impressive dataset on a large number of individuals (although see my specific comment on the number of individuals per year) and the integration of physiological requirements with estimates of energy intake is a key strength. I found the analyses to be well conducted and	We thank the reviewer for their kind words about our study and its presentation in the manuscript, and we are pleased that they believe the paper is a useful contribution to our growing understanding of variation in the responses of marine predators with different physiological requirements to climate variability and change.

could identify no major problems. I appreciate that the authors have made their code available. Overall, the manuscript is well-written and neatly presented. The figures are clear and attractive. In my opinion the results are interesting and relevant in terms of the ability of marine predators to respond flexibly to variations in environmental conditions, given that the ability of predators to respond to such variations will be an important factor in how they deal with environmental change. In contrast to lots of other work on, for example seals and seabirds, I found the thermal requirements of tuna to be an interesting element in this work which nicely illustrates the constraint of meeting physiological requirements in the environment while finding food.	
Is it a reasonable assumption that tuna are targeting the same or similar prey when their distribution is so different across time? That is, does the relationship between the measured 'heat increment of feeding' and the energy ingested hold when tuna change their distribution? I see mention of this in the discussion (with reference to Whitlock et al. 2013), but perhaps an additional sentence or two on the implications for the results could be added.	This is a great question. We acknowledge that there is some uncertainty regarding spatial and temporal patterns in HIF data owing to uncertainty about diet composition at different places and times. Unfortunately, spatially-explicit diet data is not available at the broad scales of movement seen in this study, which is needed in order to address this concern in a data-driven way. Based on the reviewer's recommendation, we have now added a paragraph about bluefin tuna diet where there is information (southern-central California only) and some implications of potential unobserved variability, and we emphasise the need to collect diet data across broad spatial scales, in order to more accurately track changes in energy consumption by marine predators in relation to climate change throughout their ranges (Lines 442 – 456).
L22: 'seasonal movements of prey resources'. To my mind this excludes the seasonal emergence and disappearance of resources (or changes in resource abundance), which may also drive migration (rather than simply being a case of prey resources moving). This statement also does not quite agree with one in the introduction (L59-61) where you do point out that movement strategies may be flexible, in response to more spatio-temporally dynamic (less predictable?) resource distributions.	We agree and have now changed this phrase to 'seasonal patterns of prey availability' to make it clearer that we are talking about phenological changes in both abundance and distribution of prey, rather than just movement of mobile prey.

L65: Replace 'seeing' with 'undergoing'	Agreed, changed
L67: Give the typical timescale of these marine heatwaves so that the contrast with the later 'century-scale climate change' is more clear.	We have now added the qualifying statement 'similar magnitudes of disruption to thermal habitats over scales of weeks to years as century-scale climate change'
L71: Perhaps make it clear that, rather than the cues degrading, there is a breakdown between the cue and the condition/state, etc. the cue is signalling. Minor semantics, but when I read 'cues that degrade' I got the impression of weakening signals rather than poor or false signals (and I think the latter is the case for ecological traps). Maybe Sherley et al. (https://doi.org/10.1016/j.cub.2016.12.054) is a better example than Gremillet et al. (2008) (same Benguela system)?	We agree, and have changed the language to be more precise: 'If foraging migrations are hard-wired to specific locations, or if animals follow cues to productivity that become unreliable under novel climate conditions...' We have also added the Sherley 2017 reference as this is an excellent example of this phenomenon.
L86: Perhaps a size/mass indication for 'large'? 'Large' would mean very different things to this journal's broad readership and I guess many of us do not know just how big tuna are!	Great point, we have now added maximum lengths and weights for mature bluefin tuna for reference here.
L92: Rephrase for clarity, please. Do you mean here that the population level is 95.5% lower than what it was prior to fishing? Or that the proportion of the population that is unfisher (not sure what that would mean) has decreased by 95.5%	We have now rephrased this to make it clearer that the stock is estimated to have been depleted to only 4.5% of its original size prior to industrial fishing.
L98: While you have used tracking data, I would say that you (arguably) have not assessed their 'migration pathway' (a longitudinal approach), but only their spatial distribution (cross-sectional approach) at discrete times. Maybe 'spatio-temporal distribution' or something like that?	We have now changed most instances of 'migration pathway' in the context of presenting out results. We have changed to 'latitudinal migration extent' to be more specific.
L117: Presumably climate variability and change are not only related with energetic costs for highly mobile animals? The winners and losers narrative is widespread. Maybe change to 'can buffer highly mobile animals against the shifting energyscape associated with...'	Yes, we have now changed this to 'shifting energy landscape'.

L132: Looking at the size distribution of individuals across years (Figure S1), it seems that individuals in 2011-2013 were very small (young) compared to other years (explained, I imagine, by small sample sizes in at least some of those years). What kind of influence do you think this (and the very small sample sizes in some years [2010, 2013], generally) has on the results?	Correct, the fish in some years were smaller on average due to differences in relative availability of size classes to tagging efforts in those years. We have attempted to address the question of how variability in size distribution affects migration extent in Figure S5. Although there is an effect of size on migration extent, this is mediated by climate such that temperature has a stronger effect on migration behaviour than size. For example, despite being below average in size relative to the size distributions over 15 years, fish in 2013 and 2015 migrated further north than fish in any other year during parts of the migration where data were available due to the temperature anomalies in those years. Our results may have been even more extreme if larger fish were tagged in those years, but the trend and inferences of our study appear robust. We have included this information in lines 302-308 of the results, to help facilitate interpretation of these issues of data variability.
L160-161: Tag-observed SST also in surface waters as for the temperature average? And the same SST product used later in the analyses?	We have now clarified that tag-observed SST was taken in the top 1m of the water column
L168-170: How were these bounds determined? Given that your manuscript is about environmental variability over 15 years, was the boundary of the California Current LME dynamic over the study? From Figure S3 I see that some movements are clearly long-distance migration outside the California Current, but what about more borderline cases? Do you think they could be responses to dynamic conditions, just as the north-south movements you focus on are?	The geographic bounds of the California Current LME are fixed, and we created the study area and grid based on a global LME spatial database. We agree with the assessment by the reviewer that the real ecological bounds of this system are likely to be dynamic in response to oceanographic and ecological conditions, and that the movements of animals including tuna in and out of this area is likely related to dynamism in these ecological bounds. However, the LME domain as a static boundary recognized by scientists and managers is a convenient way to represent this system and provides a consistent domain within which to track relative changes in the energy landscape through time.
L173: 'migration path'. See my earlier comment	We have changed this to 'latitudinal migration extent' for precision.
L175-176: Does calculating this moving average not introduce much more non-independence in the data that are used in the GAMs and GLMs?	Taking this running mean is in effect applying a preliminary smooth onto the data, which is then smoothed again by the GAMs. This will not fundamentally change the shape of these curves as it is just reducing high frequency variability that the

	models are smoothing out anyway. The purpose of the GAMs is to estimate the pattern of seasonal movements through time rather than estimate the effect of covariates, so no independence is assumed, and temporal autocorrelation is not a statistical artefact to be avoided in this case.
L176: I don't think I understand how the pairwise distances are calculated for 15 years. Are the pairwise distances for adjacent years, and then the mean is taken?	Mean pairwise distances (via a distance matrix) are calculated for each day across all 15 years, not just adjacent years. The mean of those means is ultimately presented. We have updated the methods to clarify.
L260: This result highlights my earlier question about the small individuals in some years. Were mean latitudes biased downwards in those years because only small individuals were tracked? Did this pattern of larger individuals occurring further north hold within years, or did size confound the year-to-year variation in foraging latitude? Perhaps give the GLM results here if they speak to this question.	We don't believe this is the case – 2011, 2012 and 2013 had unusually small fish relative to the mean across the time series, but by no means the lowest migration latitudes. 2013 and 2015 actually had the largest anomalous northward latitudes for the part of the year where we had data (Fig 1). While we recognize that fish size is an effect that interacts with climate variability to shape bluefin tuna migrations (as shown in Figure S5 and now discussed in lines Lines 276-281), it appears that temperature variability in the California Current is a dominant force in shaping distributions, and that this temperature variability acts on fish of all sizes. We have now discussed this more clearly in the manuscript.
L263-264: See my earlier request to clarify how this is pairwise distance is calculated.	We have now clarified this method and added additional clarifying text too.
L279-280: It would be useful to have the degree shifts as approximate km values too (as in the abstract).	Thank you, we agree and have now done this in every case where we mention climate-driven shifts in foraging latitude.
L391: Delete 'fitness', since you do not assess the fitness of these individuals as usually defined. Energetic benefits?	Changed to 'potential energetic benefits'.
Figure 1: From the truncation of the smooths in some years, I guess that in these years the data did not cover the whole year. How would this influence your results? Could the results from some years cause bias simply because they did not cover the whole year?	This is correct, variability in tagging effort over 15 years resulted in some years where tagged fish were only at large for part of the year (2002 when the program started, and 2013-2014 which had only the first and second parts of the year covered, respectively due to no tagging effort in 2013). Unfortunately, there's not much we can do to fill these gaps. We do not believe there is any fundamental bias introduced by only having fish at large during part of the year in 3/15 years, given that we do not provide annual estimates of any parameters, or explicitly compare years. The periods when we do have data in

	these years, the fish show trends in migration behaviour that are in line with the major findings that we present – namely in 2013 and 2014 when waters were anomalously warm, fish were further north than average. It is possible that the magnitude of our findings have been biased down by not having more data from these years to include in analyses but we cannot answer these questions explicitly without data. We have strived to be transparent about data availability and distribution across years, including this information in Figures S2 and S4 and now in the methods line 148-152. We have now included discussion of these points in lines Lines 276-281 of the results to aid interpretation.
Figure S4: It seems that part A of this figure is missing (and part B is not labelled as such). Also, one can figure out which smooth response is which based on their relative colours, but it would be better to label or clearly colour the two lines.	Thanks for catching this, we had mis-labelled the figure to include a panel that we no longer wish to include in supplement. We have now removed reference to this panel in the figure caption. We have also now labelled the two lines to make it clearer which represent high vs low temperatures
Reviewer 2	
Dynamic energy landscapes are critical in determining movement patterns of predators, yet incorporating energy intake and expenditure to determine such movement is challenging. The authors use energy intake (from tags) and expenditure (from temperature) to create a dynamic energy landscape in iconic bluefin tuna, and show how tuna alter their migration to optimize their landscape each year. I have only a few relatively small comments	We thank Reviewer 2 for their comments on our work, their suggestions are much appreciated and we have incorporated them into the revised manuscript.
Most of the figures show modelled data using GAMMs. Thus, it's not possible to assess whether trends are driven by outliers, the goodness-of-fit or whether the non-linearity is appropriate. I encourage the authors to add the actual data, perhaps as additional panels, to figures 1 and 3.	We have used GAMs as a means of identifying seasonal signals in the data, due to the large number of individuals and days (~65,000 data points = a lot of information to present on a single plot). In an effort to be as transparent as possible about the underlying distribution of the data and its variability, we have now included a figure in supplement showing individual panels for each year with the GAM trends from Figure 1, plus raw underlying data points so that interested readers can see this variability (Figure S4).
2. In Figure 2, why the SE's? I would have thought the SD would be more appropriate. Otherwise,	This is a good point, thanks. We have now shown SD instead of SE, so that the pattern is not solely the

you are essentially calculating $1/\sqrt{\text{sample size}}$ and so representing the amount of data you collected rather than variability.	effect of more northerly cells being more variable because they are visited less often. These figures are now in S6(A, B, C)
3. I do not believe the raw (location and other sensor) data is available. It is standard to make such data available via DRYAD or Movebank or similar.	All the data needed to conduct these analyses (tag ID, fish size, daily locations, ambient temperature, internal temperature, etc) are contained in the processed file 'hiflocs.csv' in the linked GitHub repo. The code needed to reproduce the study needs only this file, and the lead author has only used this file and not the raw data. Raw data files for all sensors for all fish over 15 years are available upon request from Prof. BA Block, we have added this to the Data Accessibility statement.
4. The calibrations of body temperature as a measure of energy intake appear to be small-scale. Over such large scales could it be more difficult to detect body temperature drops in warm waters and so the lower energy intake is partially an artefact of lower body temperature drops in warm water?	The effects of temperature on the magnitude of heat increment of feeding is accounted for in the way the models were tested and calibrated (models were laboratory calibrated between temperatures of 15 and 22°C, and the algorithm is adjusted based on ambient temperature). We have now added this to methods, line 158-169. The reviewer is correct that beyond a certain temperature and body size fish are able to conserve heat more effectively and HIFs may become harder to detect. However, we do not believe that this has biased our results in a significant way because the wild fish spent ~87% of their time in ambient temperatures that are within the range of calibration temperatures that were used to develop the heat increment of feeding models, and <5% of their time at temperatures above this range (>22°C).
5. L129. Why six days?	This is based on an assessment in the first study to apply this analysis to wild fish, where some tuna were observed not to feed for several days after release. Six days was judged to be a conservative point at which feeding activity became indistinguishable from fish that had been at large for longer periods. We have now added some more detail here and the reference to previous work to clarify.

Appendix B

Dr. Gemma Carroll
School of Aquatic and Fisheries Sciences &
NOAA Alaska Fisheries Science Center
Seattle, USA

Dear Dr. Dall,

I am writing to submit a revision of our manuscript titled "**Flexible use of a dynamic energy landscape buffers a marine predator against extreme climate variability.**"

We have addressed the editor's final remaining concerns regarding data availability. Raw sensor data from the tuna tags are available online and are downloadable via www.gtop.org and we have now added this information to the data accessibility statement. We have also included information about sources of funding that supported this work that were not included in the last version of the manuscript.

We appreciate the time and attention that went into reviewing our paper, and we are excited to publish our work in Proceedings of the Royal Society B.

Sincerely,

Gemma Carroll
